# Intercontinental Gut Microbiome Variances in IBD

**DOI:** 10.3390/ijms231810868

**Published:** 2022-09-17

**Authors:** Luis Mayorga, Gerard Serrano-Gómez, Zixuan Xie, Natalia Borruel, Chaysavanh Manichanh

**Affiliations:** 1Microbiome Lab, Vall d’Hebron Institut de Recerca (VHIR), Vall d’Hebron Hospital Universitari, Vall d’Hebron Barcelona Hospital Campus, Passeig Vall d’Hebron 119-129, 08035 Barcelona, Spain; 2Departament de Medicina, Universitat Autònoma de Barcelona, 08193 Cerdanyola del Vallès, Spain; 3Crohn’s and Colitis Attention Unit, Digestive System Service, Hospital Vall d’Hebron, 08035 Barcelona, Spain

**Keywords:** microbiome, IBD, geography

## Abstract

The development of biomarkers for inflammatory bowel disease (IBD) diagnosis would be relevant in a generalized context. However, intercontinental investigation on these microbial biomarkers remains scarce. We examined taxonomic microbiome variations in IBD using published DNA shotgun metagenomic data. For this purpose, we used sequenced data from our previous Spanish Crohn’s disease (CD) and ulcerative colitis (UC) cohort, downloaded sequence data from a Chinese CD cohort, and downloaded taxonomic and functional profiling tables from a USA CD and UC cohort. At the global level, geographical location and disease phenotype were the main explanatory covariates of microbiome variations. In healthy controls (HC) and UC, geography turned out to be the most important factor, while disease intestinal location was the most important one in CD. Disease severity correlated with lower alpha-diversity in UC but not in CD. Across geography, alpha-diversity was significantly different independently of health status, except for CD. Despite recruitment from different countries and with different disease severity scores, CD patients may harbor a very similar microbial taxonomic profile. Our study pointed out that geographic location, disease activity status, and other environmental factors are important contributing factors in microbiota changes in IBD. We therefore strongly recommend taking these factors into consideration for future IBD studies to obtain globally valid and reproducible biomarkers.

## 1. Introduction

Inflammatory bowel diseases (IBDs), which include Crohn’s disease (CD) and ulcerative colitis (UC), are multifactorial, chronic conditions of the gastrointestinal tract that have been associated with gut microbiome alterations [1,2,3,4,5,6,7,8,9]. Reduced microbial community richness and diversity, as well as reduction of beneficial microbes and expansion of a variety of potentially harmful bacterial species, have been consistently reported in IBD, particularly in CD patients. However, diet, drugs, ethnicity, geography, and a multitude of lifestyle or environmental variables may confound the interpretation and replication of microbiome studies. This is particularly problematic with small cohort size studies in which these covariates may mask the disease effect. The importance of geography and related environmental exposure has been well-illustrated in migration studies, where a strong association between microbiome diversity and migration was clearly demonstrated [10,11]. Vangay et al. [11] performed 16S and deep shotgun metagenomic sequencing on stool samples collected from individuals living in Thailand and the USA, including first- and second-generation immigrants before and after immigration. Intriguingly, USA immigration was found to be associated with significant alterations in the gut microbiome, including loss of diversity, bacterial strains, fiber degradation functions and a shift from *Prevotella* to *Bacteroides* strains.

Commonly, and quite consistently, the relative abundance of *Enterococcus*, *Fusobacterium*, *Streptococcus, Escherichia coli*, and *Ruminococcus gnavus* has been found positively associated with IBD, while that of *Faecalibacterium prausnitzii* spp., and members of *Roseburia* was negatively related to IBD [6,12,13,14,15,16,17]. However, some bacterial taxa identified as biomarkers for IBD were also found in other chronic diseases including type 2 diabetes (T2D). For instance, an enrichment in Christensenellaceae and *Escherichia coli* was linked to CD and T2D-associated dysbiosis [6,18,19], questioning the specificity of the available microbiome signatures for disease discrimination.

Based on published datasets including prospective and cross-sectional population and patient cohorts [15,19,20,21], Metwaly et al. [15] reviewed the distribution of microbial profiles in IBD and T2D across different geographical regions. The authors identified geography as a major factor in the taxonomic composition variation of the gut microbiome. Another large longitudinal intercontinental study, using the 16S approach in 531 patients with IBD from Ireland and Canada, showed that geographic location was also the major determinant of microbiome variations, although most (90.3%) of the compositional variance remains unexplained [22]. Altogether, these studies emphasized the need to take into account this variable for the identification of specific disease biomarkers. Despite the fast-growing microbiome field, studies comparing intercontinental geographical differences in IBD gut microbiome remain scarce, probably due to the presence of confounders such as environmental and host-related factors and a lack of sample processing standards.

In the present study, we aimed to investigate microbiome variations in IBD across countries using published DNA shotgun metagenomic data. Our results highlight the clinical, anthropometric, and demographic data that were commonly available and have allowed us to perform association analysis with microbiome variations to elucidate how these covariates may impact on taxonomic variations. We also emphasized the lack of standardized clinical data that has prevented us from integrating cohorts into this intercontinental study or has limited the interpretation of the results. Our findings also demonstrated that, despite recruitment from different countries and with different disease severity scores, CD patients may harbor a very similar microbial taxonomic profile.

## 2. Results

### 2.1. Participants’ Characteristics and Fecal Samples

To evaluate microbiome variation across countries, we compared our partially published Spanish IBD cohorts [23,24] with the USA [14] and Chinese [25] published cohorts. Characteristics of the cohorts are described in Table 1. For the Spanish and Chinese cohorts, we generated taxonomic and functional profiles from the shotgun metagenomic sequence reads; for the USA cohort, we downloaded these profiles from the inflammatory bowel disease multi-omics database (see Methods section). In total, 1470 fecal samples were included, 625 from patients with CD, 384 from patients with UC, and 461 from HC (or non-IBD) (Table 1).

### 2.2. Fecal Microbiome Variation Explained by Covariates in the Metadata

For the whole dataset (*n* = 1470), the most commonly reported variables from the published metadata included country, age, disease activity index, body mass index (BMI), and gender. To evaluate the impact of these covariates on microbiome variation, we applied the PERMANOVA method (adonis function) to the unweighted and weighted UniFrac distances between samples. While country and disease activity status contributed to most of the microbiome variations, the remaining covariates also showed a significant impact (Figure 1).

For the HC group, available covariates included country, age, gender, and BMI; for CD, disease localization, country, age, gender, BMI, and Harvey Bradshaw index (HBI); and for UC, country, age, gender, BMI, and colitis activity index (CAI) (Table 1, Appendix A). For HC and UC, country appeared to be the most significant explanatory variation factor, whereas for CD, disease localization (based on the Montreal classification system) explained most of the variation (Figure 1). Both CAI (in UC) and HBI (in CD) indices had a significant effect on microbiome variation, however, CAI appeared to have a higher impact than HBI. Gender, BMI, and age presented a much lower effect than country and disease. As all these covariates had a significant impact on microbiome composition, they were taken into account for the differential abundance analysis described below.

### 2.3. Geographical and Disease-Related Dysbiosis

In this study, dysbiosis is defined as a deviation from a healthy microbiome composition at the alpha- and beta-diversity levels. To assess alpha-diversity, we calculated the Chao1 and Shannon indices [14,26,27]. The Chao1 richness estimator gives weight to the low-abundant species as it only accounts for singletons and doubletons, whereas the Shannon index accounts for both the abundance and evenness of the species present. The dysbiosis score is defined as the median weighted or unweighted UniFrac distance of any given sample against a reference set built using samples from HCs [24].

One common diversity-based observation that could be drawn from the comparison of CD, UC, and HC in the three geographical cohorts was that CD displayed a significantly lower microbial diversity based on Shannon and Chao1 indices compared with HC and UC (Figure 2). UC was associated with lower richness compared with HC in both the Spanish and the USA cohorts but presented lower evenness compared with HC only in the USA cohort, which suggested the contribution of dominant taxa in UC dysbiosis in the USA cohort.

CD exhibited lower richness (Chao1 index) compared with UC in both the Spanish and the USA cohorts, but lower evenness (Shannon index) only in the Spanish cohort, which again suggested the important contribution of dominant taxa to the differences between CD and UC in the Spanish cohort (Figure 2A).

Although we observed a greater severity of both CD and UC in the USA cohort compared with the Spanish cohort (Figure 3A), we could appreciate a lower diversity only in the USA UC cohort and not in the USA CD cohort (Figure 4B,C). Furthermore, a significant negative correlation between CAI and both Shannon and Chao1 indices (Figure 3B) was found for UC, while no significant correlation was observed between HBI and CD (Figure 3C). These results indicated that disease severity was associated with diversity in UC, but not in CD.

Remarkably, we observed that, across countries, HCs from the USA were associated with a lower diversity compared with those from Spain and China (Figure 4). This finding was not explained by BMI, age or any other available covariates. It could be explained by the lack of a mechanical lysis step during the DNA extraction procedure (https://ibdmdb.org/cb/document/Sample%20Handling%20Protocols/DNA_and_RNA_Stool_Isolation_HMP2_protocol.pdf, accessed on 14 September 2022) or by environmental factors such as diet, but this latter information was not available in the published metadata. Unexpectedly, diversity was not significantly different in CD between the Spanish and USA cohort, whereas the Chinese cohort presented a lower diversity than the other two countries. Again, this difference could be related to missing covariates such as genetic as well as disease severity or diet in the Chinese metadata.

Beta-diversity analysis using weighted and unweighted UniFrac distances showed that, at the global microbial composition level, CD patients formed a separate cluster from that of HC and UC. Using the PERMANOVA test, all comparisons were significant (pval < 0.005): HC vs. CD for both weighted and unweighted UniFrac; HC vs. UC for unweighted and weighted UniFrac, CD vs. UC for both weighted and unweighted UniFrac (Appendix A), which is in agreement with previous findings [3,6]. Beta-diversity was more affected by CD localization than clinical activity (Figure 1), however, this observation could be explained by the fact that the majority of CD patients were in clinical remission.

To evaluate the degree of dysbiosis in CD and UC, we calculated the dysbiosis score based on weighted and unweighted UniFrac dissimilarities to HC, as described by Lloyd-Price et al. [14]. Based on unweighted UniFrac dysbiosis scores, as expected, CD and UC were both significantly different from HC in Spain and the USA, and CD from HC in the Chinese cohort (Figure 5). However, based on the weighted UniFrac dysbiosis scores, the differences remained significant only in the Spanish and Chinese cohorts. This latter observation could be explained by a lower alpha-diversity of HC in the USA cohort compared with the other two cohorts. Moreover, CD was different from UC in Spain based on both weighted and unweighted UniFrac dysbiosis scores, however, in the USA cohort, CD differed from UC only based on the unweighted UniFrac dysbiosis scores. This latter observation indicated that alterations of the microbiome community in UC were more related to low-abundant taxa.

To compare the degree of dysbiosis related to disease phenotype across countries, we calculated the ratio of median CD dysbiosis scores over that of HC. This analysis, using unweighted UniFrac dysbiosis scores, revealed a higher dysbiosis level in the Chinese (ratio of 1.84) followed by the Spanish (ratio of 1.79) and the USA (ratio of 1.17) cohorts (Figure 5A). This ratio, using weighted UniFrac dysbiosis scores, followed the same trend (ratio of 1.329 for China; ratio of 1.315 for Spain; ratio of 0.957 for the USA). The latter result indicated a lack of difference between CD and HC in the USA cohort (pval > 0.05).

### 2.4. Taxonomic Profile Associated with Disease Phenotype and Severity

The taxa were classified at the species level and filtered by achieving at least 0.1% of total abundance in at least 10% of the samples in order to remove outlier taxa with very low abundance and prevalence. Mixed-effects linear models, as implemented by MaAsLin2, were used to determine differences between CD, UC, and HC taxonomic profiles, as well as between countries with the same disease phenotype.

We first compared the taxonomic profile of CD (*n* = 563) with that of HC (*n* = 409), combining the Spanish and USA cohorts, taking into account covariates, such as gender, age and body mass index (BMI) included as fixed effects, while subject ID, HBI, country, and CD localization were accounted as random effects. Samples from the Chinese cohort were excluded from these analyses due to the missing CD localization data. Results showed that the most significantly depleted species in CD compared with controls were *Ruminococcus bromii* (adj. pval = 0.03) and *Ruminococcus bicirculans* (adj. pval = 0.042), while *Escherichia coli* (adj. pval = 0.03) was the most enriched species in CD (Appendix A, Figure 6, Appendix A). However, UC (*n* = 384) compared to HC (*n* = 409), with age, gender, and BMI as fixed effects, while subject ID and country were handled as random effects, did not lead to significant differences at the species level.

To reveal bacterial species associated with relapse or remission status in CD and UC, we split the CD and UC cohorts into two groups. The remission state consisted of 444 CD samples with HBI < 4 and 295 UC samples with CAI < 4 and the relapse state encompassed 119 CD samples with HBI ≥ 4 and 89 UC samples with CAI ≥ 4. In neither the CD nor UC group did disease activity status associate with specific bacterial species using MaAsLin2.

### 2.5. Taxonomic Differences across Countries and Health Status

Country-wise analysis between Spain and the USA showed that four species were differentially abundant in HC samples (*n* = 67 Spain, *n* = 342 USA) (Appendix A, MaAsLin2 results for HC/country group), eight species in UC (*n* = 65 Spain, *n* = 319 USA) (Appendix A), and no species achieved significance in CD (*n* = 57 Spain, *n* = 506 USA) (using age, gender, and BMI as fixed effects while subject ID, HBI or CAI, and country as random effects using MaAsLin2).

The relative abundance of *Fusicatenibacter saccharivorans* was significantly different in the HC group between the Spanish and USA cohorts, but did not achieve significance in the UC group, while *Agathobaculum butyriproducens*, *Coprococcus catus, Blautia wexlerae*, *Anaerostipes hadrus,* and *Eubacterium halii* were significantly enriched in the Spanish UC group, but not HC, whereas *Coprococcus comes*, *Dorea formicigenerans* and *Dorea longicatena* were significantly enriched in the both the Spanish HC and UC groups (Figure 6).

Interestingly, we did not find any species associated with disease severity for both CD (when using HBI as a possible explanatory variable) and UC (when using CAI as a possible explanatory variable) for both the Spanish and the USA cohorts.

## 3. Discussion

The relationship between microbiota and IBD pathogenesis is well established, but there is still limited information regarding the influence of geography and its related environmental impact on IBD microbiota composition. To our knowledge, this is the first study that compares gut microbiome variances in IBD patients across three continents using DNA shotgun metagenomics.

Geographical differences observed may reflect differences attributed to host genetic, dietary habits, and lifestyle (living conditions, sanitation measure, urbanization level, stress level, antibiotics used in animal agriculture, etc.) [28]. The lower alpha-diversity described in both the healthy and UC USA cohorts could be attributed to diet. Indeed, according to the United States Department of Agriculture (USDA), consumers in the EU and the United States are very different, with less consumption of fruits and vegetables in the latter population, which could lead to a lower alpha-diversity (https://www.ers.usda.gov/webdocs/outlooks/40408/30646_wrs0404f_002.pdf, accessed on 14 September 2022).

Analyzing the influence of the metadata covariates on the microbiome, we observed that geographic location was a major determinant of microbiome variation in both IBD and HC, agreeing with previous studies [22,29,30]. In a large multicenter longitudinal study on early life, Stewart et al. found that environmental factors, including geographical location and household exposures, represented important covariates of the microbiome structure. More recently, in an intercontinental microbiome study, Clooney et al. [22] showed that the presence or absence of a CD diagnosis followed by geographic location had the greatest impact on the microbiome. Moreover, He et al. [29] characterized the gut microbiome of 7009 healthy individuals from 14 districts within one Chinese province and found that, among various covariates, host location presented the strongest associations with microbiome variations. In addition to geography, in this study, we have demonstrated that disease, age, BMI and gender also have a significant impact on microbiome composition, although to a lesser extent.

With respect to dysbiosis, we have shown that disturbances in fecal microbiome composition were most pronounced in CD than in UC, which is in line with previous reports. Scanlan et al., described that the temporal stability of dominant species in CD was significantly lower than in HC [31]. Morgan et al., found that IBD population and ileal CD, in particular, were associated with a dysbiosis characterized by changes in Firmicutes and Proteobacteria phyla [4]. Lloyd-Price et al., followed 132 subjects for one year to generate integrated longitudinal molecular profiles of host and microbial activity during disease [14]. Samples from participants with CD or UC were overrepresented in the dysbiotic set, with 24.3% and 11.6% of their samples classified as dysbiotic, respectively. In addition, our dysbiosis findings in CD were observed regardless of geographic location and disease severity, which has not been reported elsewhere.

In the Spanish cohort, we confirmed our previous findings [24] that CD was associated with a significant reduction in diversity, based on both Shannon and Chao1 indices, compared with HC and UC, while UC did not have a lower diversity compared to HC, suggesting that the dysbiosis score might be higher in CD than in UC. In the USA cohort, based on the Shannon index, we observed differences in alpha-diversity only between UC and HC and between CD and HC, but not between CD and UC. These findings could be explained by the greater disease severity in the USA cohort than in the Spanish one. In HC, the USA cohort was associated with lower diversity compared with the other two cohorts. These findings were not explained by the available published metadata, suggesting that other unavailable factors, such as diet or ethnicity, could play a role in this observation.

Based on disease phenotype and combining the Spanish and USA cohorts, *Ruminococcus bromii and Ruminococcus bicirculans* were the significantly most depleted microbial species in CD compared with HC, while *Escherichia coli* was the most enriched species. Only the latter is in agreement with other reports [32]. Interestingly, Fang et al. [33] who performed a strain level analysis of *E. coli* in CD, found that the strain identified by the metagenomic approach was similar to known pathogenic Adherent-Invasive *E. coli* (AEIC) strains. Furthermore, *Ruminococcus bromii* has been shown to have the ability to degrade dietary-resistant starches while *Ruminococcus bicirculans* has the capacity to utilize plant glucans. These two bacterial species, missing in CD, could be key players in the metabolism of a plant-based diet and maintenance of gut homeostasis. In UC, no significant differences were observed at the species level. The fact that no species achieved significance in CD in our country-wise analysis suggested that the development of CD might lead to a convergence of this composition’s profile, and this finding could open the possibility of identifying a “core” CD microbiome as a universal biomarker.

Neither in the CD nor the UC group did disease activity status associate with specific bacteria. This finding could be explained by the fact that other variables such as BMI and age had a greater impact on the microbiome variation than the disease severity scores. In this sense, we found reports with divergent results that suggested that clinical activity had an impact. In a large pediatric CD cohort, Gevers et al. [12] described an axis defined by an increased abundance of bacteria including Enterobacteriaceae, Pasteurellacaea, Veillonellaceae, and Fusobacteriaceae, and decreased abundance of Erysipelotrichales, Bacteroidales, and Clostridiales that correlated strongly with disease status. Recently, Clooney et al., in a longitudinal study, demonstrated that the greatest changes in microbiota composition were linked to transitions across active and inactive phases of the disease, even though specific taxa were not associated with different activity states [22]. This was only possible when using intra-individual ratios of each taxon between two consecutive time points, but not when using taxa from single time points. Similarly, Lloyd-Price et al., showed that periods of disease activity were also marked by increased temporal variability, with characteristic taxonomic, functional, and biochemical shifts [14].

Globally, *Coprococcus comes*, *Dorea formicigenerans,* and *Dorea longicatena* were shared among countries but enriched in the Spanish cohort. These species were previously found to be linked to overweight/obesity in some reports [34,35] and to correlate with metabolic markers in others [36].

The use of shotgun metagenomic can be viewed as an advantage over 16S rRNA sequencing. Since 16S rRNA sequencing is a cost-effective method for taxonomic profiling, it has been widely used to investigate the association between gut microbiota and IBD [37]. However, this approach presents several pitfalls, as it is biased by the lack of truly universal primers for the PCR amplification and the variability of the rRNA operon copy number throughout the bacterial kingdom. Moreover, due to the short size of the 16S rRNA gene (about 300 bp usually sequenced), this approach leads to a taxonomic resolution mainly up to the genus level. Instead, the shotgun metagenomic approach, which consists of a random DNA sequencing from the complete content of a clinical sample, relies, in this study, on single copy marker gene databases for taxonomic profiling. This method avoids all the above-mentioned limitations and allows classification of known and unknown microorganisms at the species level.

An important limitation of our study is that it analyzed fecal and not mucosal microbiome. It is well established that feces are just a proxy for the intestinal mucosa, the actual site where interactions between bacteria and host immunity occur. However, applying shotgun metagenomics on this type of sample is still challenging as it is considered low biomass sample and contains more human cells than microbial cells. Therefore, the sequencing depth should be much higher for mucosal than for fecal samples. Our study points out other limitations when attempting to perform intercontinental research on IBD-specific biomarkers. Among these limitations, we reported the lack of standardized clinical data and the unavailability of important covariates, including dietary data, which has been previously suggested as a potential risk factor for IBD [38]. In addition, the metadata lacked information regarding other valuable covariates such as ethnicity, disease behavior, location in UC, calprotectin, and endoscopic activity. Altogether, missing data decreases the margin of analysis of confounders responsible for microbiome variations. Indeed, a non-negligible number of patients had to be discarded from the analysis due to the lack of a standard clinical activity index throughout the cohorts. For instance, the Chinese cohort could not be used in the analysis of the association between taxonomic profile and patients’ characteristics as it lacked HBI and disease location using the Montreal classification systems, as reported in the other studies. Moreover, we had to exclude an entire Belgium cohort as it did not use the same index for disease activity (Rutgeerts score instead of HBI). Another important point to consider was the lack of standardized DNA extraction methods, including the reporting of the use of a mechanical or chemical procedure across countries, which may limit interpretations concerning the observed geographical impact. Indeed, this extraction step in microbiome studies has been shown to be a possible contributor to the microbiome variation [39]. Therefore, micro bio-biomarkers identified based on the USA cohort, which did not perform additional mechanical lysis, should be taken with caution. Finally, sequence analysis methods should also be homogenized in particular in terms of taxonomic and functional database versions, as the interpretation of taxonomic and functional profiling may depend on the mapping rate.

In conclusion, geographic location, disease activity status, and other environmental factors appear to influence microbiota changes in IBD. In the case of CD, we are inclined to believe that these factors have a lower impact on microbiome composition, even though dysbiosis was greater in CD than in UC. We strongly recommend taking into consideration all possible environmental and host factors, as well as the implementation of standardized extraction methods, when conducting microbiota studies in order to obtain globally valid and reproducible findings. Finally, as IBD is a multifactorial disorder, to acquire better insights into the ethology of IBD, future ideal study design should integrate mucosal shotgun metaomic techniques, host genetics and clinical phenotypes, lifestyle, and dietary habits.

## 4. Materials and Methods

### 4.1. Collection of Metagenomic Data

Shotgun metagenomic sequencing data generated from 329 public human fecal samples were downloaded from NCBI SRA [40] (https://www.ncbi.nlm.nih.gov/sra/, accessed on 14 September 2022). The 329 public human metagenomic data were derived from two unique bioprojects, one of which was published in our previous studies (PRJEB1220, 212 samples, Spanish cohort), and the other was from the PRJEB15371 bioproject (117 samples, Chinese cohort). The taxonomic and functional profiling of HMP (NIH Human Microbiome Project, https://doi.org/10.1038/s41586-019-1238-8, accessed on 14 September 2022), which includes 1,639 samples, were downloaded from the IBDMDB (The Inflammatory Bowel Disease Multi-Omics Database, https://ibdmdb.org/, accessed on 14 September 2022) [14]. The metadata for all the human metagenomic data can be found in Appendix A and contains available information such as country, disease localization, gender, age, BMI, dysbiosis index, and severity scores. The taxonomic profiles from the HMP data were obtained using the MetaPhlAn 3.0 tool and provided relative abundances [41].

### 4.2. Filtering of the Datasets

All metagenomic datasets were filtered based on metadata information to perform a homogenized meta-analysis. We removed samples from (1) HC without body-mass indexes (BMI), age, or gender information; (2) from CD without disease localization defined by the Montreal classification method [42], Harvey-Bradshaw index (HBI) [43] (Harvey and Bradshaw 1980), gender, age or BMI information; and (3) from ulcerative colitis (UC) without Colitis Activity Index (CAI) [44], gender, age or BMI information. The taxonomic profiling tables were further filtered by criteria of species with less than 0.1% abundance [2] and samples with zero species abundance were removed. The Chinese cohort was included in the alpha-diversity, beta-diversity, and dysbiosis analyses, but was excluded from the analysis of multivariate association between covariates and microbiome features using MaAsLin2 because information on disease localization was not based on the Montreal classification method, in contrast to the other two cohorts, and HBI information was not available (Appendix A).

### 4.3. Upstream Sequence Analysis: Quality Control, Decontamination and Profiling

For the Spanish and Chinese cohorts, the KneadData v0.7.4 pipeline (https://huttenhower.sph.harvard.edu/kneaddata, accessed on 14 September 2022) was used to trim the reads with lengths below 50% of the total input read length and to remove reads that mapped to the human genome. Taxonomic and functional profiling was performed by the HumanN3 pipeline [41] against the built-in database of Metaphlan3 and the UniRef90 database with default parameters.

### 4.4. Statistical Analyses

Statistical analyses were performed using R (version 4.0.3). Alpha-diversity was assessed using Shannon [27] and Chao1 metrics [26] indices, while beta-diversity was estimated by calculating the weighted and unweighted UniFrac distances between samples [45]. The dysbiosis score was calculated as previously defined [14,24], using weighted and unweighted UniFrac distances. Healthy controls from each country were set as the reference group for each cohort. Alpha-diversity and dysbiosis scores between groups were compared using the Mann-Whitney U test. The contribution of clinical, anthropometric, and demographic factors to the effect size was estimated using permutational analysis of variance (PERMANOVA), as implemented in the adonis2 function of the vegan R package (https://cran.r-project.org/web/packages/vegan/index.html, accessed on 14 September 2022) using weighted and unweighted UniFrac distance metrics.

UniFrac distances were calculated as implemented in the phyloseq R package [46]. Differential abundance analysis of the taxonomic profiles was performed using the R version of the MaAsLin2 tool [47], which implements linear mixed-effects models that are useful for multivariable association discovery in population-scale microbiome studies.

## Figures and Tables

**Figure 1 ijms-23-10868-f001:**
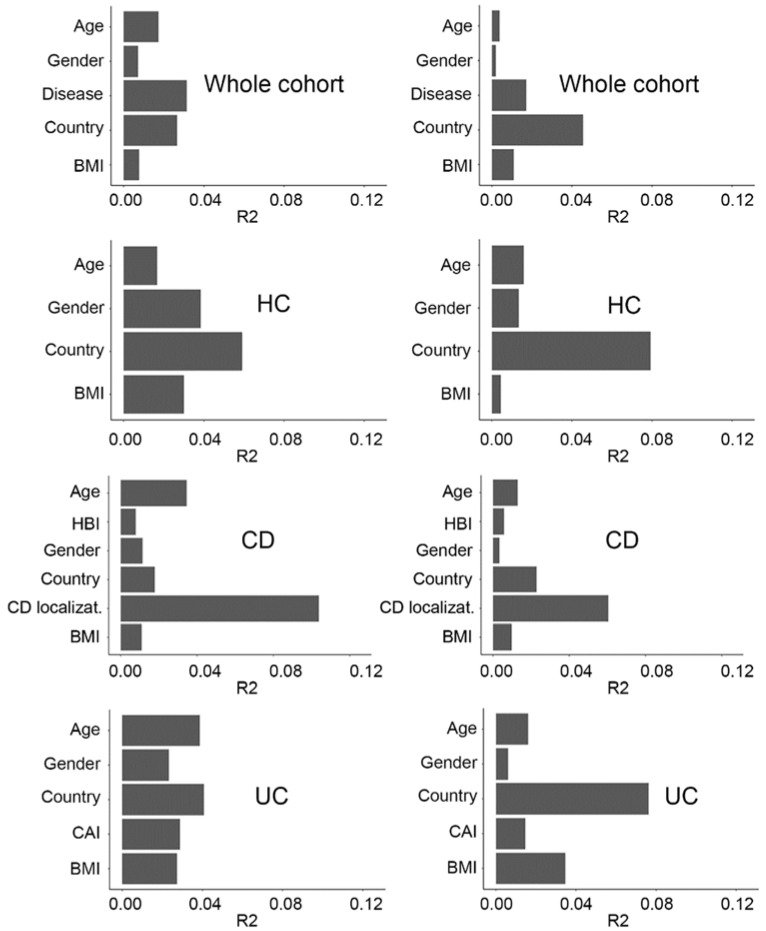
Covariates explaining fecal microbiome variance (PERMANOVA with unweighted (left plot)/weighted (right plot) UniFrac distances, 999 permutations). All covariates have a significant impact on microbiome variation (q < 0.05), except for BMI in HC (weighted UniFrac), gender in CD and UC (weighted UniFrac). Whole cohort (*n* = 1408), HC (*n* = 461), UC (*n* = 384), CD (*n* = 563).

**Figure 2 ijms-23-10868-f002:**
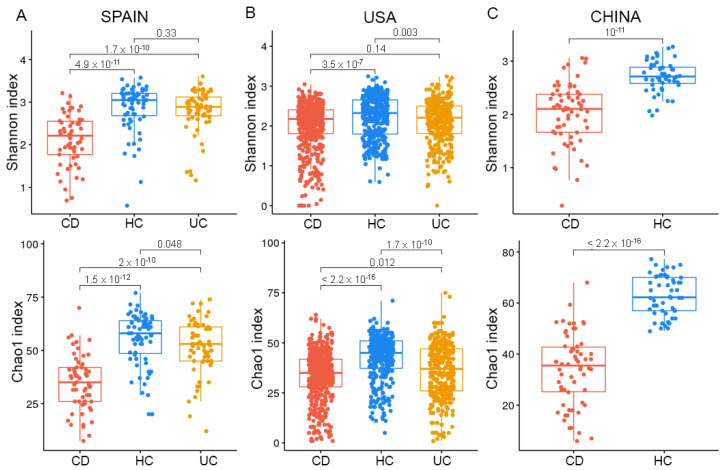
Alpha-diversity across cohorts. (**A**) In the Spanish cohort, Shannon and Chao1 indices showed a significant decrease in both microbial richness and evenness in CD. (**B**) In the USA cohort, the differences in Shannon and Chao1 indices were maintained when comparing CD with HC, but not when comparing CD with UC. (**C**) The Chinese cohort also showed decreased richness and evenness in CD compared with HC.

**Figure 3 ijms-23-10868-f003:**
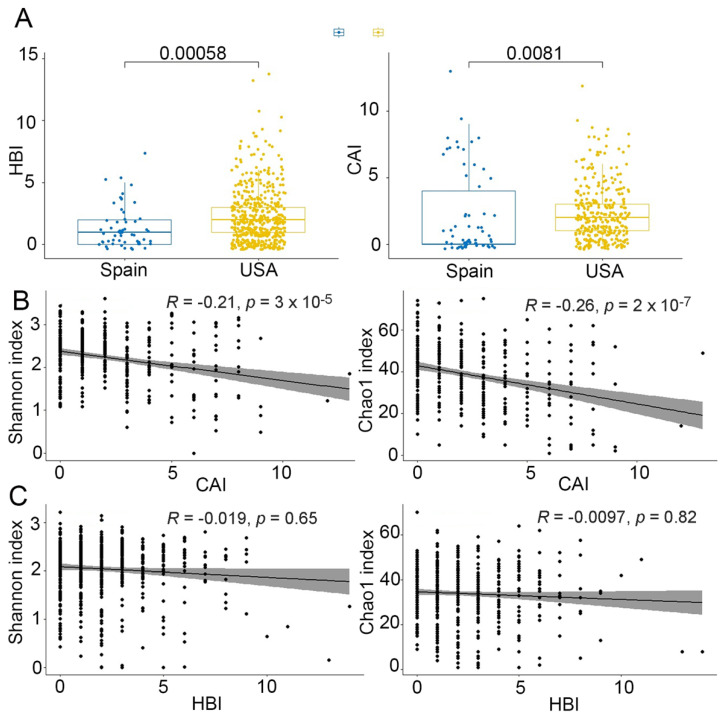
Disease activity score in CD (HBI, Harvey-Bradshaw index) and UC (CAI, colitis activity score). (**A**) HBI and CAI were both higher in the USA than in the Spanish cohort (Mann Whitney’s U test). (**B**) Correlation between alpha-diversity (Shannon and Chao1 indices) and CAI (Spearman correlation test). rho = −0.019, *p* = 0.65 for the Shannon index and rho = −0.0097, *p* = 0.82 for the Chao1 index. (**C**) Correlation between alpha-diversity (Shannon and Chao1 indices) and HBI (Spearman correlation test). rho = −021, *p* = 3 × 10^−5^ for the Shannon index and rho = −0.26, *p* = 2 × 10^−7^ for the Chao1 index.

**Figure 4 ijms-23-10868-f004:**
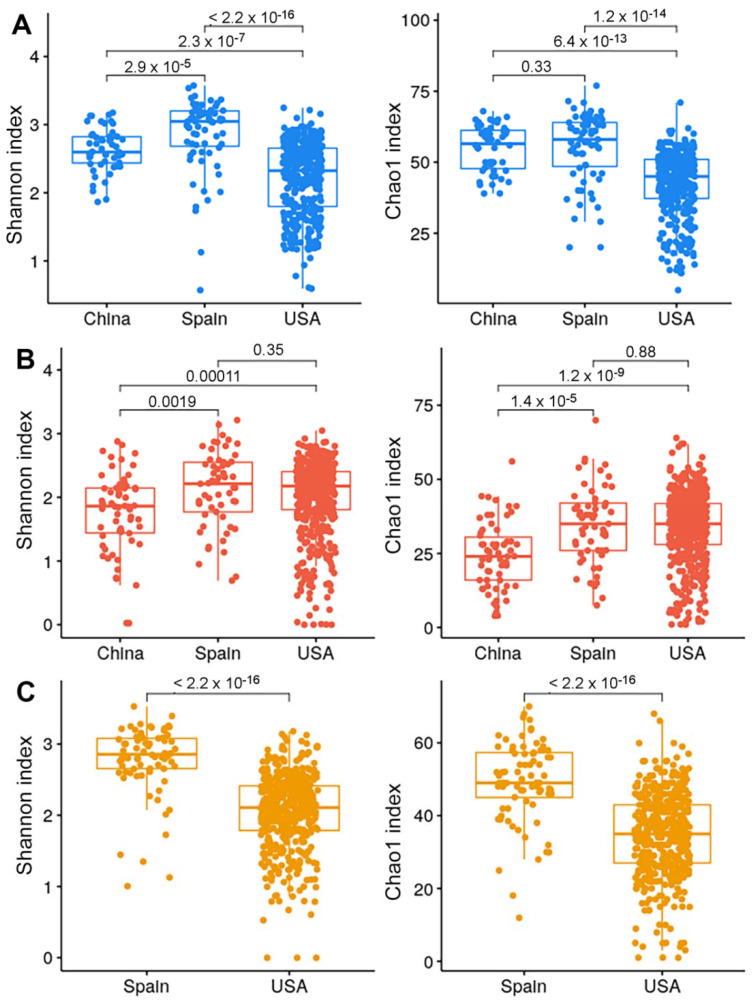
Alpha-diversity based on health status across countries. Shannon and Chao1 indices in HC (**A**), CD (**B**), and UC (**C**) (Mann-Whitney’s U test).

**Figure 5 ijms-23-10868-f005:**
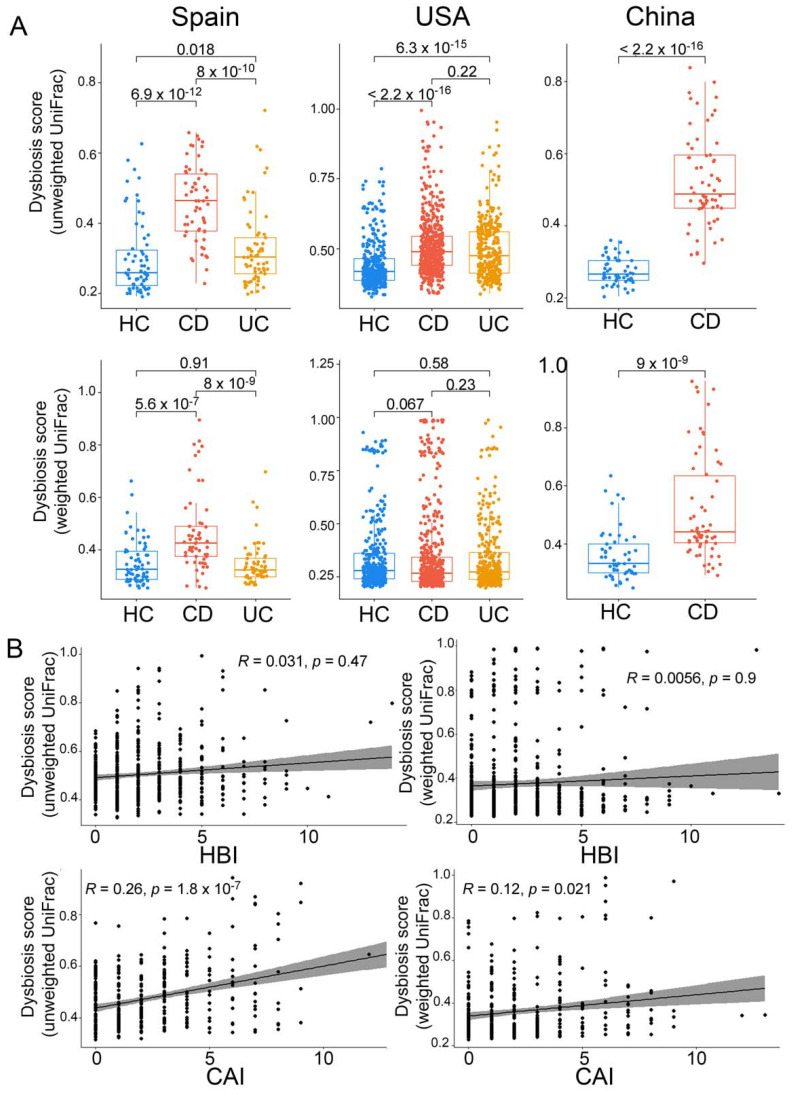
Dysbiosis scores and correlation with disease severity. (**A**) Dysbiosis scores across cohorts per country (Wilcoxon test). (**B**) Correlation between dysbiosis score and disease activity index (Spearman correlation test). rho = 0.031, *p* = 0.47 for unweighted UniFrac and rho = 0.0056, *p* = 0.9 for weighted UniFrac of CD (HBI, Harvey Bradshaw index); rho = 0.26, *p* = 1.8 × 10^−7^ for unweighted UniFrac and rho = 0.12, *p* = 0.021 for weighted UniFrac of UC (CAI, colitis activity index).

**Figure 6 ijms-23-10868-f006:**
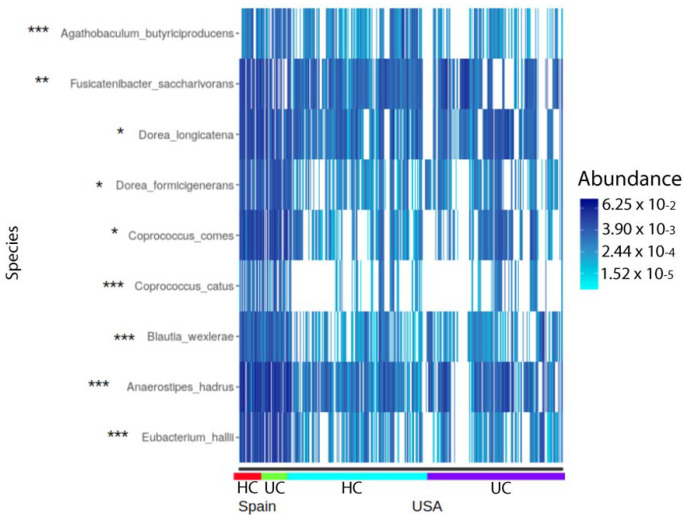
Common differentially abundant (DA) species between Spanish and USA cohorts in the HC and UC groups. * DA in both HC and UC. ** DA only in HC. *** DA only in UC.

**Table 1 ijms-23-10868-t001:** Participants’ characteristics.

	United States				Spain				China			
Type of Cohort	CD	UC	Non-IBD	Total	CD	UC	Non-IBD	Total	CD	UC	Non-IBD	Total
**Number of subjects**	37	22	20	79	33	33	67	133	62	NA	52	114
**Number of samples**	506	319	342	1167	57	65	67	189	62	NA	52	114
**CD location (*n*)** **L1/L1+L4** **L2/L2+L4** **L3/L3+L4**	109/45 58/23 128/143	NA	NA	NA	18/2 0/0 35/2	NA	NA	NA	NA	NA	NA	NA
**Gender** (% female)	39.3	65.8	50.9	50.0	66.7	72.3	56.7	65.1	25.8	NA	5.8	5.8
**Age** (mean ± SD)	22.5 ± 14.2	29.2 ± 19.5	27.5 ± 19.5	27.1 ± 17.7	35.8 ± 11.8	40.1 ± 10.5	43.1 ± 15.9	27.5±16.9	28.4 ± 8.1	NA	20.5 ± 7.8	20.5 ± 7.8
**BMI** (mean ± SD)	23.3 ± 7.5	23.0 ± 6.8	23.4 ± 7.0	23.3 ± 6.8	22.4 ± 4.0	23.6 ± 4.1	25.0 ± 4.3	23.2±6.7	18.6 ± 2.8	NA	21.4 ± 3.3	21.4 ± 3.3

L1: terminal ileum, L2: colon, L3: ileocolon, L4: upper GIT (gastrointestinal tract).

## Data Availability

All information regarding data used in this study can be found under “Collection of Metagenomic Data” in the Method section.

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
