# Peer review of "Intercontinental Gut Microbiome Variances in IBD"

_ijms, 2022, doi:10.3390/ijms231810868_

Round 1
Reviewer 1 Report (New Reviewer)
I found it interesting that microbiome variations in IBD patients and even in healthy cohorts are correlated to geography, but such a word as geography or environments is too vague to imagine concrete agencies. Any speculation is welcome in Discussion.
minor comments
comment 1. 1a. The legend of Table 1 is simply "Participants' characteristics." First of all, it is not clear why the content in Table 1 is duplicated in the upper and lower half. 1b. The legend should include what CD localization implies and what are meant by L1, L2, L3 etc. 1c. The legend may also contain the definitions of HBI and CAI. Unless otherwise they should be provided somewhere in the text.
comment 2. In Figure 2, I assume that the eight left plots (unweighted) be labeled in the same way as the eight right plots (weighted). It is also helpful to make clear distinction between the weighted and unweighted.
comment 3. The reference style is somewhat obscure (as to a comma after authors) and inconsistent (as to the presence or absence of publication years), which are exemplified as Vangay et al., [11] in line 42, Price et al., 2019 [14] in line 183, and almost all other references.
Author Response
Dear reviewer,
We appreciate your time for considering and commenting on our work. As requested, we have modified the manuscript to address your recommendations and criticisms, and are now providing this revised and improved version for your consideration. Please our point-by-point responses to your comments below.
--------
I found it interesting that microbiome variations in IBD patients and even in healthy cohorts are correlated to geography, but such a word as geography or environments is too vague to imagine concrete agencies. Any speculation is welcome in Discussion.
Response: As suggested, we extended our discussion on the possible mechanism underlying the geographical differences.
“Geographical differences observed may reflect differences attributed to host genetic, dietary habits, and lifestyle (living conditions, sanitation measure, urbanisation level, stress level, antibiotics used in animal agriculture, etc) [28]. The lower alpha-diversity described in both the healthy and UC USA cohorts could be attributed to diet. According to the United States Department of Agriculture (USDA), consumers in the EU and the United States are very different, with less consumption of fruits and vegetables in the latter population, which could lead to a lower alpha-diversity (https://www.ers.usda.gov/webdocs/outlooks/40408/30646_wrs0404f_002.pdf).”
minor comments
comment 1. 1a. The legend of Table 1 is simply "Participants' characteristics." First of all, it is not clear why the content in Table 1 is duplicated in the upper and lower half.
Response: we believe that the track changes system shows both the tables we removed and the new one we added during our first revision. We have now accepted the changes and in this way the two tables should not appear in the pdf version of the manuscript.
1b. The legend should include what CD localization implies and what are meant by L1, L2, L3 etc. 1c.
Response: Thank you for this suggestion. We have now added the meaning of these abbreviations below the table.
“L1: terminal ileum, L2: colon, L3: ileocolon, L4: upper GIT (gastrointestinal tract)”
The legend may also contain the definitions of HBI and CAI. Unless otherwise they should be provided somewhere in the text.
Response: We spelled out these abbreviations in lines 434 and 436, respectively.
comment 2. In Figure 2, I assume that the eight left plots (unweighted) be labeled in the same way as the eight right plots (weighted). It is also helpful to make clear distinction between the weighted and unweighted.
Response: We assume that the reviewer is talking about Figure 1. Yes, this way of labelling was a request from a previous reviewer.
comment 3. The reference style is somewhat obscure (as to a comma after authors) and inconsistent (as to the presence or absence of publication years), which are exemplified as Vangay et al., [11] in line 42, Price et al., 2019 [14] in line 183, and almost all other references.
Response: Thank you for this remark. We have now homogenised the reference style throughout the MS.
Reviewer 2 Report (New Reviewer)
This is an interesting paper in investigation of general biomarkers of the gut microbiome of IBD across three continents. The study confirmed that geographic location was a major determinant of microbiome variation in health and IBD, a well-observed phenomenon by many publications using 16 S rRNA sequencing method.
As geographic location may reflect variance in lifestyle, ethnicity, diet, gene-environment interactions, more detailed study in future are needed to address possible explanations for this geographic location-gut microbiome variation to provide more insight into the etiology of IBD. The author should add more discussion about this point and possible ecological processes behind the strong geographic effect.
The author should also discuss the advantage of using shotgun metagenomic sequencing compared with 16 S rRNA sequencing. Otherwise, the study can be viewed as a simple replicate from 16 S rRNA sequencing studies and only add more geographic locations. More information from the metagenomic sequencing data such as the KEGG pathways may provide more insights into functional changes in IBD gut microbiota across continents.
Please discuss more about the limitations of this study and future prospects in IBD research. The major drawback is that the study used fecal samples rather than mucosal tissues. It is well established that environment dominates over host genetics in shaping human gut microbiota. However, whole exome sequencing analyses also find gene-microbiota interactions in IBD which suggests that genetic variances associated with microbiota also affect the immune system. Local gut epithelium and immune cells microenvironment are important factor besides gut microbiome and host genetics. Therefore environmental factors, multi-omics of mucosal gut microbiome, and host genetics can be integrated together through trans-disciplinary studies for future IBD research to provide better insights into the etiology of IBD and for personalized medicine in IBD. As many problems present in finding universal intercontinental biomarkers, more mechanistic study using local healthy controls at regional level would be more meaningful.
Author Response
Dear reviewer,
We appreciate your time for considering and commenting on our work. As requested, we have modified the manuscript to address your recommendations and criticisms, and are now providing this revised and improved version for your consideration. Please our point-by-point responses to your comments below.
--------------
This is an interesting paper in investigation of general biomarkers of the gut microbiome of IBD across three continents. The study confirmed that geographic location was a major determinant of microbiome variation in health and IBD, a well-observed phenomenon by many publications using 16 S rRNA sequencing method. As geographic location may reflect variance in lifestyle, ethnicity, diet, gene-environment interactions, more detailed study in future are needed to address possible explanations for this geographic location-gut microbiome variation to provide more insight into the etiology of IBD. The author should add more discussion about this point and possible ecological processes behind the strong geographic effect.
Response: As suggested, we extended our discussion on the possible mechanism underlying the geographical differences. This suggestion was also proposed by the reviewer #1.
“Geographical differences observed may reflect differences attributed to host genetic, dietary habits, and lifestyle (living conditions, sanitation measure, urbanisation level, stress level, antibiotics used in animal agriculture, etc) [28]. For instance, the lower alpha-diversity described in both the healthy and UC USA cohorts could be attributed to diet. Indeed, according to the United States Department of Agriculture (USDA), consumers in the EU and the United States are very different, with less consumption of fruits and vegetables in the latter population, which could lead to a lower alpha-diversity (https://www.ers.usda.gov/webdocs/outlooks/40408/30646_wrs0404f_002.pdf).”
The author should also discuss the advantage of using shotgun metagenomic sequencing compared with 16 S rRNA sequencing. Otherwise, the study can be viewed as a simple replicate from 16 S rRNA sequencing studies and only add more geographic locations. More information from the metagenomic sequencing data such as the KEGG pathways may provide more insights into functional changes in IBD gut microbiota across continents.
Response: We agree with this suggestion and have now extended the advantage of doing shotgun metagenomic over 16S rRNA sequencing. The functional analysis of the datasets was not planned (as agreed with the editors) in this manuscript and will be included in another article in a near future.
“The use of shotgun metagenomic can be viewed as an advantage over 16S rRNA sequencing. Since 16S rRNA sequencing is a cost-effective method for taxonomic profiling, it has been widely used to investigate the association between gut microbiota and IBD [37]. However, this approach presents several pitfalls, as it is biased by the lack of truly universal primers for the PCR amplification and the variability of the rRNA operon copy number throughout the bacterial kingdom. Moreover, due to the short size of the 16S rRNA gene (about 300 bp usually sequenced), this approach leads to a taxonomic resolution mainly up to the genus level. Instead, the shotgun metagenomic approach, which consists of a random DNA sequencing from the complete content of a clinical sample, relies on single copy marker gene databases for taxonomic profiling. This method avoids all the above-mentioned limitations and allows classification of known and unknown microorganisms at the species level.”
Please discuss more about the limitations of this study and future prospects in IBD research. The major drawback is that the study used fecal samples rather than mucosal tissues.
Response: We agree and have now added this limitation in the discussion section.
“An important limitation of our study is that it analysed fecal and not mucosal microbiome. It is well established that feces are just a proxy for the intestinal mucosa, the actual site where interactions between bacteria and host immunity occur. However, applying shotgun metagenomics on this type of sample is still challenging as it is considered low biomass sample and contains more human cells than microbial cells. Therefore, the sequencing depth should be much higher for mucosal than for fecal samples.”
It is well established that environment dominates over host genetics in shaping human gut microbiota. However, whole exome sequencing analyses also find gene-microbiota interactions in IBD which suggests that genetic variances associated with microbiota also affect the immune system. Local gut epithelium and immune cells microenvironment are important factor besides gut microbiome and host genetics. Therefore environmental factors, multi-omics of mucosal gut microbiome, and host genetics can be integrated together through trans-disciplinary studies for future IBD research to provide better insights into the etiology of IBD and for personalized medicine in IBD.
Response: we appreciate this comment and have added the following text at the end of the manuscript.
“Finally, as IBD is a multifactorial disorder, to get better insights into the ethology of IBD, future ideal study design should integrate mucosal shotgun metaomic techniques, host genetics and clinical phenotypes, lifestyle, and dietary habits.”
As many problems present in finding universal intercontinental biomarkers, more mechanistic study using local healthy controls at regional level would be more meaningful.
Response: yes, we agree with this statement. In the present work, we chose only studies with healthy controls from the same region than the patient cohorts.
This manuscript is a resubmission of an earlier submission. The following is a list of the peer review reports and author responses from that submission.
Round 1
Reviewer 1 Report
The manuscript entitled ‘Intercontinental gut microbiome variances in IBD’ by Mayorga L., et al., compares alpha-diversity as well as in dysbiosis scores in correlation with disease severity using microbial sequence data obtained from fecal samples of patients, who were originally recruited as three separate cohorts. The cohort data used in the paper were from the Spanish IBD cohorts first presented in 2017 (Pascal V, et al. Gut 2017;66:813–822), a study including patients from five academic medical centers in USA (Lloyd-Price J, et al. Nature 2019; 569:655-662), and a study including patients from a single institute in China (He Q, et al. Gigascience 2017; 6:1-11). The authors of the current manuscript have not been not collaborators of the latter two studies, and the sequence data of the latter two studies are available from the deposited database.
Line 22 Conclusions: The descriptions are what the authors would like to point out, and not exactly the conclusion of the study.
Line 48: I do not understand for what reason the authors think “unfortunately”.
Line 54: Please re-consider the word “analyzed” since reference 15 is a review paper.
Line 77: I think reference 23 should be 27.
Lines 256, 261-263: A selection bias cannot be excluded when choosing sample data to be included in the current analysis. Reference 14 has already been published, and it has demonstrated that there was pronounced dysbiosis in CD. This particular data from reference 14 was selected to be used in for analysis in the current paper, and the authors of the current paper concludes identical findings to that of the reference 14, that there is dysbiosis in CD.
If their own Spanish data had the same trend as the reference 14, it may be considered to refer to the previously published paper by Lloyd-Price, instead of incorporating the previously published data into their own Spanish data for analysis.
Line 283: Please specify the reference for ‘country-wise analysis’.
Lines 69-70, 76-77, 329: Please show ethical approval for all participants when using patient information parameters.
Author Response
We appreciate your time spent considering and commenting on our work. As requested, we have modified the manuscript to address your recommendations and criticisms, and are now providing this revised and improved version for your consideration. Please, find our point-by-point responses to your comments below.
Line 22 Conclusions: The descriptions are what the authors would like to point out, and not exactly the conclusion of the study.
Response: We appreciate this remark from the reviewer and have made the following change:
Conclusions: “Our study pointed out that geographic location, disease activity status, and other environmental factors are important contributing factors in microbiota changes in IBD. We therefore strongly recommend taking these factors into consideration for future IBD studies to obtain globally valid and reproducible biomarkers.”
Line 48: I do not understand for what reason the authors think “unfortunately”.
Response: We used the term “unfortunately” to denote that some of the biomarkers were not specific for discriminating IBDs but were identified for T2D as well, which questions the specificity of the biomarkers for a disease of interest. To clarify our thoughts, we have modified the text as follows:
“However, some bacterial taxa identified as biomarkers for IBD were also found in other chronic diseases including type 2 diabetes (T2D).”
Line 54: Please re-consider the word “analyzed” since reference 15 is a review paper.
Response: We thank the reviewer for mentioning this mistake:
“Based on published datasets including prospective and cross-sectional population and patient cohorts [15, 19-21], Metwaly et al., [15] reviewed the distribution of microbial profiles in IBD and T2D across different geographical regions. “
Line 77: I think reference 23 should be 27.
Response: The reviewer is right, the dataset we used in this manuscript was also used in reference 27, however, reference 23 is the original paper, where the reader can find the sequence accession number. We have, therefore, added both references.
Lines 256, 261-263: A selection bias cannot be excluded when choosing sample data to be included in the current analysis. Reference 14 has already been published, and it has demonstrated that there was pronounced dysbiosis in CD. This particular data from reference 14 was selected to be used in for analysis in the current paper, and the authors of the current paper concludes identical findings to that of the reference 14, that there is dysbiosis in CD. If their own Spanish data had the same trend as the reference 14, it may be considered to refer to the previously published paper by Lloyd-Price, instead of incorporating the previously published data into their own Spanish data for analysis.
Response: We agree that using published data could always bias the analyses. In this study we re-analyzed either the sequence data or compositional tables from three different countries. We believe that it would be very difficult to design a study and collect samples from large IBD cohorts from different countries with a reasonable timing and taking advantage of published data makes sense in this context. One of the main objectives of this study was to encourage the inclusion of standardized clinical and demographic data, as well as the use of standardized sample extraction protocols in the design of future studies either in the context of IBD or any other disorders.
Line 283: Please specify the reference for ‘country-wise analysis’.
Response: We meant in “our” country-wise analysis. The fact that no species achieved significance in CD in our country-wise analysis suggested that the development of CD might lead to a convergence of this composition’s profile, and this finding could open the possibility of identifying a “core” CD microbiome as a universal biomarker.
We have now clarified this aspect in the manuscript.
Lines 69-70, 76-77, 329: Please show ethical approval for all participants when using patient information parameters.
Response: Since the three studies made their patient information parameters public, according to the ethical committee of our hospital, we do not need to show ethical approval again.
Reviewer 2 Report
This subjet is interesting. In fact a better known and comprehension of differents evolutions of microbiota in function of geographical localization most permet a better prevension and better treatment adapted of population specificity.
The authors installe correctly the subject in their introduction.
The cohorts used are disequilibrate. In fact USA cohort are most elevated, Spain and China have less patients than USA. It could be explain for Spain by dimension of the pays but not for China cohort. And China cohorte demonstred absence of patients informations.
Il seems difficult to have a real comparison of cohorts.
In figure 1 the elements information are not in the same order. This complique the comparison of results. There is a necessirity to reclass them
The results explaining the difference between the cohorts are clear and well designed for the other figures.
In conclusion, the study compare 3 important continents for the IBD pathology.
However, Chinese patients are less numerous and the European continent is represented by only one country. This continent necesite the inclusion of a other North pays and Est pays to be really representative.
In this article it’s demonstrated that microbiote composition and variation are very importantes ind the CD pathology.
The authors say there are a relation between pathology severity and geographical localization. This fact is not describe now. This vision of pathology is interesting.
The authors show a modification of the composition of the microbiota according to the pathology observed and a difference according to the location. This difference may be due, as the authors state, to variations in the treatment of the samples, which makes comparison difficult.
Furthermore, a correlation between geographical location, microbiota composition and severity are not sufficiently developed. This is a pity because it is probably the most innovative information.
It is therefore necessary to complete in an important way the data obtained with the cohorts and to put much more in visibility the relation severity/location/microbiota.
Author Response
We appreciate your time spent considering and commenting on our work. As requested, we have modified the manuscript to address your recommendations and criticisms, and are now providing this revised and improved version for your consideration. Please, find our point-by-point responses to your comments below.
This subjet is interesting. In fact a better known and comprehension of differents evolutions of microbiota in function of geographical localization most permet a better prevension and better treatment adapted of population specificity.
The authors installe correctly the subject in their introduction.
Response: We appreciate the reviewer’s comment.
The cohorts used are disequilibrate. In fact USA cohort are most elevated, Spain and China have less patients than USA. It could be explain for Spain by dimension of the pays but not for China cohort. And China cohorte demonstred absence of patients informations.
Il seems difficult to have a real comparison of cohorts.
Response: We agree with this remark. However, despite differences in numbers of samples in each cohort, and “Despite recruitment from different countries and with a different disease severity index, CD patients may harbor a very similar microbial taxonomic profile.” For UC patients, we may need larger cohorts as the reviewer suggested. Therefore, one of the main objectives of this study was to encourage the inclusion of standardized clinical and demographic data, as well as the use of standardized sample extraction protocols in the design of future studies either in the context of IBD or any other disorders.
In figure 1 the elements information are not in the same order. This complique the comparison of results. There is a necessirity to reclass them
Response: As requested, we have reordered the variable names such that they now appear in a similar order in each plot.
The results explaining the difference between the cohorts are clear and well designed for the other figures.
Response: Thank you for this remark.
In conclusion, the study compare 3 important continents for the IBD pathology.
However, Chinese patients are less numerous and the European continent is represented by only one country. This continent necesite the inclusion of a other North pays and Est pays to be really representative.
Response: We agree that adding more cohorts from each continent would be needed. To the best of our knowledge, we did not find additional studies available that could be included in this study as they did not include these criteria: IBD adult patients with healthy adult controls, available shotgun metagenomic sequence data, available metadata with clinical information and from the North and East of Europe. The only study we recently found when revising this manuscript was a study initiated by Hu et al., published in 2021 (PMID: 32651235). Unfortunately, we could not include this cohort in the current version of our manuscript as the authors performed the sequence analysis using HUMAnN2 (V.0.6.1), which uses MetaPhlan 2.0, a computational tool for profiling the composition of microbial communities. Including this cohort would imply a re-analysis of the sequence dataset using HUMAnN3, which makes use of MetaPhlAn 3.0, an updated version of MetaPhlan 2.0. In this way we would homogenize the analysis in terms of bioinformatics methods, as we did for the three cohorts currently included in this study. However, doing so, would require an additional six months of computing time and therefore would not be performed in a timely fashion. We definitely consider including this cohort in another project that would integrate many more cohorts from different regions of the world, as suggested by the reviewer.
In this article it’s demonstrated that microbiote composition and variation are very importantes ind the CD pathology.
The authors say there are a relation between pathology severity and geographical localization. This fact is not describe now. This vision of pathology is interesting.
Response: We appreciate the reviewer’s comments and have now modified the text as follows:
Line 137: “Interestingly, although we observed a greater severity of both CD and UC in the USA cohort compared with the Spanish cohort (Figure 3A), we can appreciate a lower diversity only in the USA UC cohort and not in the USA CD cohort (Figure 4BC).”
The authors show a modification of the composition of the microbiota according to the pathology observed and a difference according to the location. This difference may be due, as the authors state, to variations in the treatment of the samples, which makes comparison difficult.
Response: We understand the reviewer’s concern. We have now discussed this limitation further in the discussion section.
“Another important point to consider was the lack of standardized DNA extraction methods, including the reporting of the use of a mechanical or chemical procedure across countries, which may limit interpretations concerning the observed geographical impact. Indeed, this extraction step in microbiome studies has been shown to be a possible contributor to microbiome variation [36]. Therefore, micro bio-biomarkers identified based on the USA cohort, which did not perform additional mechanical lysis, should be taken with caution. Finally, sequence analysis methods should also be homogenized in particular in terms of the version of the taxonomic and functional databases, as the interpretation of taxonomic and functional profiling may depend on the mapping rate.”
Furthermore, a correlation between geographical location, microbiota composition and severity are not sufficiently developed. This is a pity because it is probably the most innovative information.
It is therefore necessary to complete in an important way the data obtained with the cohorts and to put much more in visibility the relation severity/location/microbiota.
Response: We agree with the reviewer that correlating severity, geography, and microbiota would be the ideal objective in this study. However, in an attempt in doing so (i.e., comparing the microbiome during relapse and remission), we encountered various obstacles: 1). The Chinese cohort did not contain severity indices; 2). The extraction method of the USA was different than the two cohorts. 3). The number of samples with very severe disease (i.e., relapse) in the Spanish cohort was very low (n=8 for CD and n=17 for UC, considered relapsed samples). Therefore, when performing MaAsLin2 analysis, we did not encounter significant results, which could be attributed to confounding factors or missing data and samples.
Reviewer 3 Report
The article must be organized following the standard structure:
1. Introduction
2. Materials and Methods
3. Results
4. Discussions
5. Conclusions
In the Discussions chapter you must explain with more details the limits and the strengths. Further, you must compare your results with other studies, using the same parameters for comparison.
Author Response
We appreciate your time spent considering and commenting on our work. As requested, we have modified the manuscript to address your recommendations and criticisms, and are now providing this revised and improved version for your consideration. Please, find our point-by-point responses to your comments below.
Comments and Suggestions for Authors
The article must be organized following the standard structure:
- Introduction
- Materials and Methods
- Results
- Discussions
- Conclusions
Response: We have structured the manuscript by putting Materials and Methods after Discussion as was recommended in the guidelines of the journal.
“Research manuscript sections: Introduction, Results, Discussion, Materials and Methods, Conclusions (optional).”
In the Discussions chapter you must explain with more details the limits and the strengths. Further, you must compare your results with other studies, using the same parameters for comparison.
Response: As requested by the reviewer, we have now improved several parts of our discussion.
“...Based on disease phenotype and combining the Spanish and USA cohorts, Ruminococcus bromii and Ruminococcus bicirculans were the significantly depleted microbial species in CD compared with HC, while Escherichia coli was the most enriched species. Only the latter is in agreement with other reports [31]. Interestingly, Fang et al., (PMID: 30425690) who performed a strain level analysis of E. coli in CD, found that the strain identified by the metagenomic approach was similar to known pathogenic Adherent-Invasive E. coli (AEIC) strains. Furthermore, Ruminococcus bromii has been shown to have the ability to degrade dietary resistant starches while Ruminococcus bicirculans has the capacity to utilize plant glucans. These two bacterial species, missing in CD, could be key players in the metabolism of plant-based diet and maintenance of gut homeostasis…”
“... Indeed, this extraction step in microbiome studies has been shown to be a possible contributor to microbiome variation [36]. Therefore, micro bio-biomarkers identified based on the USA cohort, which did not perform additional mechanical lysis, should be taken with caution. Finally, sequence analysis methods should also be homogenized in particular in terms of the version of the taxonomic and functional databases, as the interpretation of taxonomic and functional profiling may depend on the mapping rate…”
Round 2
Reviewer 1 Report
The authors have revised, or else have explained, the specific points questioned previously from the reviewer.